# An Assessment of the Lactic Acid-Producing Potential of Bacterial Strains Isolated from Food Waste

Henriette Peace Uwamahoro [1,2,*] , Fuyu Li [1,2], Arbindra Timilsina [1] , Binbin Liu [1] , Xinzhen Wang [1] and Yinping Tian [1]

1   Hebei Key Laboratory of Soil Ecology, Center for Agricultural Resources Research, Institute of Genetics and Developmental Biology, Chinese Academy of Sciences, Shijiazhuang 050021, China; lifuyu@ms.sjziam.ac.cn (F.L.); arbintms@sjziam.ac.cn (A.T.); binbinliu@sjziam.ac.cn (B.L.); xzwang@sjziam.ac.cn (X.W.); tianyinping@sjziam.ac.cn (Y.T.)
2   University of Chinese Academy of Sciences, Beijing 100049, China
*   Correspondence: peace.hope90@gmail.com; Tel.: +86-18103115474

**Abstract:** Lactic acid (LA) is widely used in many industries as a crucial starting material in food products, bio-based materials, and biodegradable polymers. The goals of this research were to isolate LA bacteria from food wastes, assess their potential for LA production, and study their growth characteristics. In this study, six bacterial strains were isolated from food waste and identified using 16S rRNA gene sequencing; namely, *Weissella viridescens* WJ39, *Leuconostoc lactis* YS33, *Leuconostoc citreum* KD42, *Leuconostoc mesenteroides* VN60, *Macrococcus caseolyticus* FCI29, and *Weissella confusa* RG41. *W. viridescens* WJ39 showed the highest potential for lactic acid production (17.56 g $L^{-1}$day$^{-1}$), and the lowest potential was found in *L. lactis* YS33 (14.09 g $L^{-1}$day$^{-1}$). There were significant differences ($p < 0.05$) in the LA production rates among *Weissella* spp., *Leuconostoc* spp., and *Macrococcus* spp. Moreover, dramatic differences in growth rate were observed among the six strains. *W. viridescens* WJ39 exhibited the highest growth rate (0.80 h$^{-1}$), while *M. caseolyticus* FCI29 exhibited the lowest growth rate (0.57 h$^{-1}$). *W. viridescens* WJ39 also exhibited lactic acid production (at a rate around 2 g $L^{-1}$day$^{-1}$) in a lab incubation experiment with food waste as a nutrient source. The draft genome of *W. viridescens* WJ39 with 16 contigs was constructed with an N50 of 215217 bp. The genome size was approximately 1.54 Mb, with a GC content of 41%. A *hicD* gene, known to catalyze the conversion of pyruvate to D-lactate, was discovered in the genome. This study illustrated the potential for the production of lactic acid from food waste with lactic acid bacteria.

**Keywords:** lactic acid bacteria; food waste; isolation; *Weissella viridescens*; growth rate

## 1. Introduction

Globally, a large amount of food is wasted every year. Food waste decomposition generates significant amounts of toxins and foul odors, such as ammonia ($NH_3$) and hydrogen sulfide ($H_2S$) [1]. Many methods are used for food waste management, including landfills and anaerobic digestion [2]. Currently, fermentation is being developed as an alternative treatment for waste management with organic acid production, including lactic acid [3,4]. Lactic acid (LA) is widely utilized in the processing of commodities such as cosmetics, food, textiles, and surfactants [5]. LA is generally produced by chemical synthesis or fermentation. The latter offers advantages over chemical synthesis due to the low cost of raw materials, low energy consumption, and production of pure lactic acid [6].

LA is a naturally occurring organic acid that can be used in a wide variety of industries, such as the cosmetic, chemical, pharmaceutical, and, most recently, the medical industries [6]. In the chemical industry, it can be used in applications such as neutralization, cleaning, green solvents, and releasing agents [7]. In the pharmaceutical industry, LA is used as an intravascular solution, dialysis solution, mineral preparation, and controlled drug delivery system, while in the cosmetic industry, it is used as a component of

moisturizers, anti-acne treatments, and anti-aging formulations, as well as skin lightening agents [7]. LA is also applied as a building block of polylactic acid (PLA), a biocompatible aliphatic polyester, and is biodegradable [8]. PLA is typically produced through the direct polycondensation of LA and ring-opening polymerization of the lactic acid cyclic dimer, known as lactide [9]. PLA has numerous biomedical, orthopedic, and dental applications. PLA is commonly used to manufacture fixation devices such as pins, screws, darts, and washers, and it is used extensively in reconstructive surgeries such as mandibular joint reconstruction, facelifts, finger, thoracic, leg, hand, and toe fractures, soft and hard tissue fixations, alignment of osteochondral ligament reconstruction procedures and bone fragments, hyaline cartilage fixation, and meniscus repair [10]. In addition, PLA is an important product for bio-based and circular economy [11]. Therefore, its global market demand has significantly increased in recent years, with a compound annual growth rate (CAGR) of 18.7% from 2019 to 2025 [5]. Worldwide, the three largest consumers of lactic acid markets are the United States, followed by China and Western Europe [12]. The global LA market size was valued at USD 2.7 billion in 2020 and is expected to expand at 8.0% CAGR from 2021 to 2028 [13].

The fermentation of lactic acid bacteria (LAB) through carbohydrate metabolism produces LA as the metabolic end-product [14]. Lactic acid bacteria (LAB) are used in fermentation processes [15]. LAB are Gram-positive, and nonspore-forming cocci or rods [16]. They are found in a wide range of habitats, including fermented dairy products, meat, and vegetables as well as in human and animal gastrointestinal and urogenital tracts [17]. LAB may also be found in sewage, dung, wastes, milk, and soil [18]. LAB, including *Lactobacillus reuteri* I2, *Pediococcus acidilactici* I5, *P. acidilactici* I8, *P. acidilactici* c3, *P. pentosaceus* I13, and *Enterococcus faecium* c14, can be isolated from broiler chickens [19]. Notably, LAB is also found in fermented vegetables, meat, and cereals [20]. For example, LAB such as *Lactobacillus acidophilus*, *Lactobacillus plantarum*, *Lactobacillus delbrueckii*, and *Lactobacillus casei* can be obtained from the carbohydrate-based fermentation product retted cassava (a wet-paste made from fermented cassava) [21]. To grow these bacteria to high levels in the laboratory, De Man, Rogosa, and Sharpe agar (MRS) medium is usually used [22]. Both MRS agar and MRS broth, consisting of peptone and dextrose, provide nitrogen, carbon, and other nutrients required for the optimum growth of LAB [23]. In addition, polysorbate 80, acetate, magnesium, and manganese provide nutrients for the growth of a variety of lactobacilli [24]. LAB are primarily used as starter cultures and probiotics. The use of starter cultures is a prerequisite for introducing modern industrial technology, which is most advanced in the dairy field, followed by meat technology and sourdough production, wineries, and probiotics. LAB are used in many food products and are applied in food fermentations [25]. LAB mainly play a role in producing LA as a product during metabolic activity. LAB also possess therapeutic properties that are vital for human health enhancement [26]. Furthermore, LAB can produce antimicrobial agents, vitamins, enzymes, and exopolysaccharides and are a source of low-calorie sweeteners, ethanol, and acetic acid in addition to LA [27].

*Bacillus coagulans* A534 is reported to have the potential to produce optically pure L-lactic acid (L-LA) [28]. *Bacillus* species, including *Bacillus coagulans*, *Bacillus licheniformis*, and *Bacillus subtilis*, are reported as potential agents for LA production [29]. Furthermore, two species of the genera *Geobacillus* (*G. stearothermophilus*) and *Halolactibacillus* (*H. halophilus*) are reported to be L-LA producers [30]. *Bacillus* species are Gram-positive, rod-shaped and aerobic endospore-forming bacteria and are found in diverse environments, such as soil and clay, dust, rocks, aquatic environments, on vegetation, in the gastrointestinal tract, and in food [31]. Moreover, *Bacillus* species, apart from LA, produce antimicrobial compounds that can be utilized for medical, food, or agricultural applications [31].

The reasons for increasing food waste around the world are consumer attitudes, values, and behaviors towards food while preparing and serving more. However, when not used in time, it could boost the rise of food waste [32]. In the European Union, consumption generates the highest amount of food waste (65%) while distribution generates the least

amount (5%) [33]. Moreover, based on food waste in the rural households of 28 Chinese provinces, 1.67% of food is wasted per household per day in rural China [34]. Furthermore, urbanization is the most powerful driving force, resulting in the gradual expansion of the supply chain to meet the food needs of city dwellers [35]. In addition, changes in dietary composition per household are linked to an increase in food waste [33]. Bacteria can successfully degrade the starch, protein, fats, and cellulose found in food waste [36]. Carbohydrates, lipids, and proteins are common components of food waste [37]. In general, starchy materials such as corn, rice, maize, potato, wheat barley, and cassava are good substrates for LA production [4]. Therefore, food waste and its constituents could be suitable substrates for the growth of bacteria (LAB and *Bacillus* strains). More investigations are needed to isolate and identify beneficial bacteria and their potential for lactic acid production. Thus, the aims of this study were (i) to isolate bacterial strains from food wastes and (ii) to determine the growth rate of the isolated bacteria and assess the lactic acid production potential of the isolated bacteria. This approach would be beneficial for the safe utilization of the tremendous amount of food waste generated daily while producing LA, which has many industrial applications.

## 2. Materials and Methods

### 2.1. Food Waste Collection

Food waste was collected from the litter bins in the canteen at the Center for Agricultural Resources Research, Institute of Genetics and Developmental Biology of CAS, located in Shijiazhuang district, Hebei province (latitude 38.0931, and longitude 114.652). An average of 200 people eat regularly in this area per meal. Food waste was collected at 2 p.m. after the end of serving and before decomposing had begun. Food waste consisted of eggs, meat, eggshells, carrots, green vegetables, rice, and bread. Other impurities (e.g., tissue papers, plastic bottles, etc.) were removed from the food waste by hand.

### 2.2. Bacterial Isolation and Culture Conditions

Ten grams of food waste were weighed and placed into flasks filled with 90 mL of sterile water. The sample suspension was prepared, maintained at 30 °C, and shaken at 160 rpm for 30 min. The supernatant was serially diluted 10-fold in a range of $10^{-1}$ to $10^{-6}$ [38]. Then, microbial strains were isolated by spreading 100 μL of the dilutions into LB agar medium (tryptone 1%, yeast power 0.5%, NaCl 0.5%, agar 15%, $H_2O$ 1000 mL) and De Man, Rogosa (MRS) agar medium that was composed of 5 g beef extract, 10 g peptone, 20 g glucose, 4 g yeast extract, 1 mL tween-80, 2 g dipotassium phosphate, 5 g sodium acetate, 2 g triammonium citrate, 0.05 g manganese sulfate, 0.2 g magnesium sulfate, and 15 g agar. Bacteria were incubated in aerobic conditions at 28 °C for 24 h with three replicates, and media without inoculation were used as controls in triplicate [24]. Colonies were picked randomly, and pure colonies were obtained according to morphological differences in size, color, margin, and shape using an aseptic technique to prevent contamination [39]. To prepare the stock for further analysis, a single colony was added to MRS and LB broth depending on the isolating medium used for each strain. All strain cultures were stored at −80 °C with 40% glycerol solution [40].

### 2.3. Molecular Identification

The 16S rRNA gene sequence was selected for species identification as it is sufficiently variable to distinguish bacterial species while also being sufficiently conserved to enable primer design. The sequences for numerous bacterial species are available in the NCBI GenBank. The amplification of the 16S rRNA gene was carried out using the primer pair 27F (5-AGAGTTTGATCCTGGCTCAG-3) and 1492R (5-CTACGGCTACCTTGTTACGA-3). A total of 1.0 μL of genomic DNA (20 ng/μL) were added to a 50.0 μL PCR containing 5.0 μL of 10× Buffer (containing 2.5 mM $Mg^{2+}$), 1.0 μL of Taq polymerase (5 u/μL), 1.0 μL of dNTPs (10 mM), 1.5 μL of 27F primer (10 μM), 1.5 μL of 1492R primer (10 μM), and 39.0 μL of ddH₂O. The reaction involved an initial denaturation at 95 °C for 5 min, followed

by 35 cycles of denaturation at 95 °C for 30 s, annealing at 58 °C for 30 s, and extension at 72 °C for 1 min 30 s, with a final extension step at 72 °C for 7 min. The AxyPrep DNA gel recovery kit (Axygen Scientific, Inc., Union City, CA, USA) was used to recover the PCR products from agarose gel. Sanger sequencing using the purified PCR products of each strain was performed on the ABI3730-XL sequencer (ThermoFisher Scientific, Waltham, MA, USA), and Blastn was then used to compare the resultant sequence with the data in the NCBI 16S database to determine species identity. A phylogenetic tree was constructed for the isolates with the neighbor-joining method using MEGA X software [41].

### 2.4. Staining of the Isolated Bacteria

All six bacteria were cultured in MRS broth for 24 h at 28 °C. Cultured cells were fixed on slides using heat. Next, for 1 min, culture cells crystal violet was added to the culture cell section, and slides were briefly rinsed under running tap water to remove excess crystal violet. Gram iodine mordant was applied for 1 min to the culture cell section and briefly washed in tap water. To eliminate any non-specific crystal violet staining, a gram decolorizer solvent (acetone) was applied to the slides for 20 s and then quickly rinsed under running tap water. In addition, safranin was added for about 1 min and washed with water until clear. Finally, using a light microscope (×100), we examined the outcomes of the staining technique under oil immersion [42,43].

### 2.5. Fermentation Conditions and Media

The pH of the medium was adjusted to 6.2 before sterilization in an autoclave for De Man–Rogosa–Sharpe (MRS) preparation. Then, six pure strains from −80 °C were propagated in medium [44]. The preculture medium was incubated at 28 °C between 12–24 h based on the bacterial growth and was left undisturbed in the rotary shakers until it attained an optical density (*OD*) of 0.5 at the exponential growth stage of the bacteria. Then, a 10% (*v/v*) culture medium with $3.9 \times 10^9$ CFU/mL for each strain was transferred to a new sterilized medium for each strain. Three replicates for each bacteria and control were used and incubated in aerobic conditions for 24 h at 28 °C with 150 revolutions per min (rpm) [4].

The bacterial strain identified as producing the highest amount of LA in MRS was selected for testing its potential for LA production in food waste. Food waste was concurrently blended to reduce its size to the range of 0.2–1.0 mm to facilitate fermentation [45]. First, the pure strain from −80 °C was activated in MRS broth at 28 °C for 12 h to reach the exponential stage before being transferred to the food waste. Three replicates and controls (without inoculum) were used. Then, the pH of the food waste was adjusted to 6.2 before autoclaving, and 5 mL of culture was inoculated in 45 g of sterilized food waste. Last, the cultures were incubated at maximum room temperature (28 °C) for six days [46].

### 2.6. Construction of the Calibration Curve

A total of 1.2 g of LA with a known concentration (89%, ρ = 1.2 g/mL) was diluted with distilled water in a 10-mL volumetric flask. Then, a series of LA solutions were prepared using 2-fold dilutions. Iron (III) chloride (0.3 g) was added to a volumetric flask of 100 mL distilled water for a solution of iron (III) chloride. Then, 100 µL of the LA solution were added to 4 mL of iron (III) chloride and stirred, and the procedure was repeated for all diluted LA solutions. The absorbance of the mixture was measured at 390 nm using a UV-6100S spectrophotometer. Finally, a calibration curve showing the relationship between the concentration of LA (g/mL) and absorbance was plotted (Figure A1) [47].

## 2.7. Lactic Acid Determination

The culture was incubated for 24 h and centrifuged at 10,000 rpm at 4 °C for 15 min. The supernatant was diluted 10-fold with deionized water for each sample [46]. Then, a suspension of 100 µL was added to 4 mL of iron (III) chloride solution and mixed well. Next, the LA content (absorbance) was measured using a UV-6100S spectrophotometer at 390 nm [45]. Finally, the unknown concentration of LA was calculated based on a calibration curve, as shown in Figure A1.

## 2.8. Bacterial Growth Rate and Growth Curve

The growth of a bacterial culture is defined as an increase in the number of bacteria in a population rather than an increase in the size of individual cells. The growth of a bacterial population occurs geometrically or exponentially with each division cycle [48]. A growth curve of the strains in MRS broth was constructed from three replicates and controls. Then, 100-µL cultures at the exponential stage were inoculated into 250 mL for 24 h at 28 °C with 150 rpm [49]. Cell density (absorbance) was measured with a UV-6100 spectrophotometer at 600 nm (*OD* 600 nm) at approximately 1 h intervals for up to 15 h [50]. The exponential stage was used to determine µ from the natural optical density logarithm of two data points versus the time corresponding to the two-point *OD* data [51]. In the following step, the growth rate of the bacterial isolated was calculated by Equation (1).

$$\mu = \frac{\ln OD_2 - \ln OD_1}{t_2 - t_1} \tag{1}$$

## 2.9. Draft-Genome Sequencing of Weissella viridescens WJ39

The inoculum of the *Weissella viridescens* WJ39 isolate, originally isolated from food waste, was cryopreserved in 50% glycerol at −80 °C. *Weissella viridescens* WJ39 was cultured in MRS broth at 28 °C for 12 h to reach the exponential stage. After DNA extraction, a library of paired-end 2 × 150 bp was constructed and sequenced on an Illumina NovaSeq sequencing platform. The adaptor sequences were removed using AdapterRemoval v (2.2.2) software [52]. FastQC was used to quality control the data of the raw reads. The genome size was estimated using SOAPec (v2.0) software [53]. The K-mer was counted using Jellyfish v (2.0.0) software [54]. A5-MiSeq [55] and SPAdes [56] were used to de novo assemble the sequencing data. We utilized CRISPRCasFinder to predict forward repeats and spacers of a genome [57].

## 2.10. Statistics

The lactic acid generated and growth rate achieved by the isolated bacteria were analyzed by one-way analysis of variance (ANOVA) to check whether there were significant differences among isolates. Three replicates and a control were used in this study. According to Duncan's multiple range test, $p < 0.05$ was considered significant. All phylogenetic trees were generated using MEGA X (10.0) software.

## 3. Results

### 3.1. Identification of Isolated Strains

Six bacterial strains, VN60, FCI29, RG41, YS33, KD42, and WJ39, were isolated and identified from food waste. VN60 and FCI29 were isolated in LB medium and subcultured in MRS medium, while RG41, YS33, KD42, and WJ39 were isolated in MRS medium and subcultured in MRS medium. Figure 1 show the growth and morphologies of the colonies for all six strains during subculture on MRS agar at 28 °C for 24 h. Gram stain enables the identification of Gram-positive bacteria (Figure 2). The 16S rRNA gene of the isolates was subjected to basic alignment local tool (BLAST) analysis against the National Center for Biotechnology (NCBI) database. Strains VN60, FCI29, RG41, YS33, KD42, and WJ39, were identified as *Leuconostoc mesenteroides*, *Macrococcus caseolyticus*, *Weissella confusa*, *Leuconostoc lactis*, *Leuconostoc citreum*, and *Weissella viridescens*, respectively. Phylogenetic analysis based

on the 16S rRNA gene showed that five of the bacteria belong to the order *Lactobacillales*, while FCI29 belongs to *Bacillales*. In addition, the phylogenetic tree (Figure 3) also revealed that *Macrococcus caseolyticus* FCI29 does not have a close relationship with the other five isolated bacteria. The 16S rRNA gene sequences of the six isolates (VN60, FCI29, RG41, YS33, KD42, and WJ39) were deposited in NCBI under accession numbers ON387328, ON387327, ON387326, ON387331, ON387330, and ON387329, respectively.

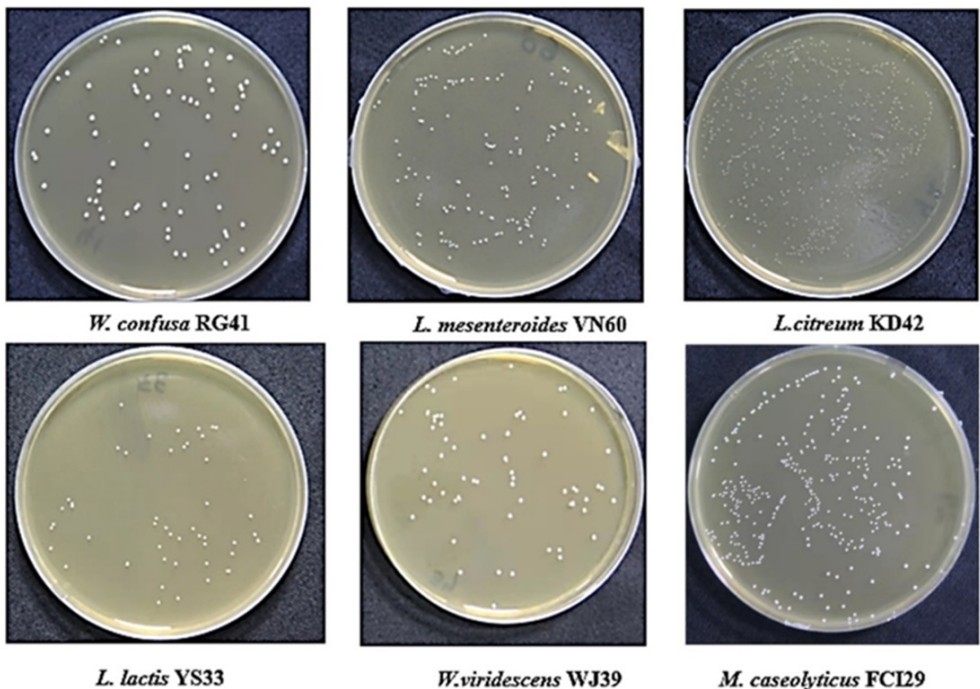

**Figure 1.** The six isolates that were grown on MRS agar at 28 °C for 24 h.

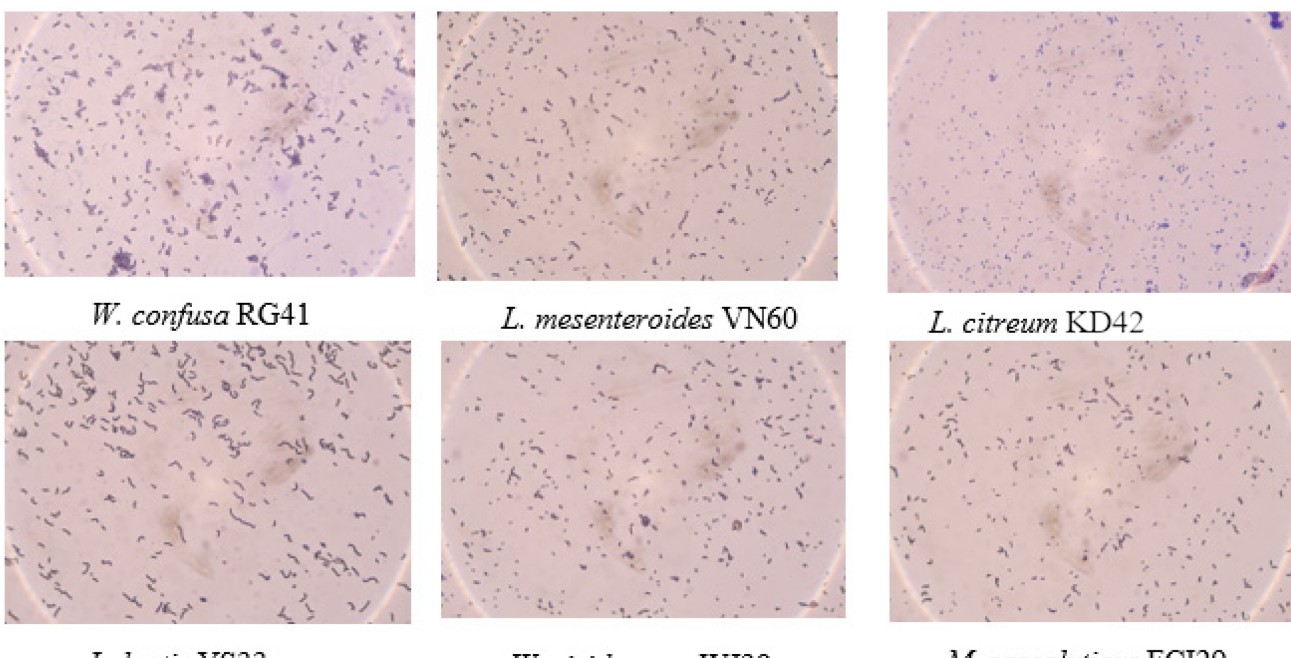

**Figure 2.** Gram staining of the six bacterial strains.

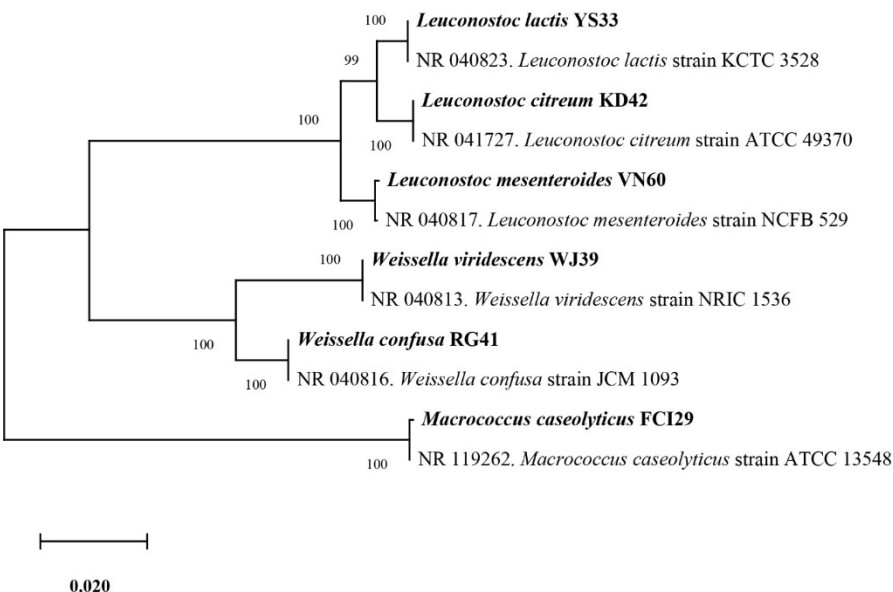

**Figure 3.** Phylogenetic tree based on the sequencing results of the 16S rRNA gene shows the relationships between all six isolated bacteria.

### 3.2. Growth Rates of the Isolated Strains

Bacteria isolated from food waste displayed growth rate values (μ) of 0.80 h$^{-1}$, 0.79 h$^{-1}$, 0.68 h$^{-1}$, 0.66 h$^{-1}$, 0.66 h$^{-1}$, and 0.57 h$^{-1}$ for *W. viridescens* WJ39, *Weissella confusa* RG41, *Leuconostoc mesenteroides* VN60, *Leuconostoc citreum* KD42, *Leuconostoc lactis* YS33, and *Macrococcus caseolyticus* FCI29, respectively. Our statistical results showed significant differences in the growth rates of the six bacteria ($p < 0.05$). According to our findings, the species from the *Weissella* genus exhibited a high growth rate, while the species from the *Macrococcus* genus showed a lower growth rate. Additionally, all bacteria reached an exponential stage within 15 h, and the progressive changes in optical density in the six bacterial species over time (h) are represented in Figure 4.

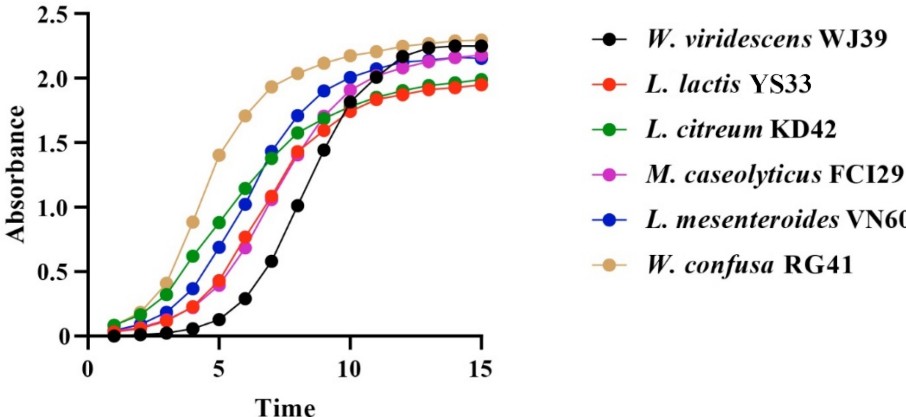

**Figure 4.** Changes in *OD* per unit time in the six identified bacterial strains at 15 h.

### 3.3. Potential for Lactic Acid Production of the Isolated Strains

The bacterial isolates exhibited the potential for LA production in MRS broth, as illustrated in Figure 5. The LA production rates from *W. viridescens* WJ39, *W. confusa* RG41, *L. mesenteroides* VN60, *L. mesenteroides* VN60, *L. citreum* KD42, and *L. lactis* YS33 were 17.56 g L$^{-1}$day$^{-1}$, 17.18 g L$^{-1}$day$^{-1}$, 15.84 g L$^{-1}$day$^{-1}$, 15.37 g L$^{-1}$day$^{-1}$, 14.15 g L$^{-1}$day$^{-1}$, and 14.09 g L$^{-1}$day$^{-1}$, respectively. We found that *W. viridescens* WJ39 produced the highest amount of lactic acid, and *L. lactis* YS33 produced the lowest amount of lactic acid in the MRS medium. According to Duncan's multiple range test, there were significant

differences ($p < 0.05$) in LA production among the *Weissella* spp., *Leuconostoc* spp., and *Macrococcus* spp. The pH range of all strains reached 4.5 for all bacteria after 24 h, where the initial pH was adjusted to 6.2. In addition, *W. viridescens* WJ39 produced an LA amount (12.46 g L$^{-1}$) within six days, which is around 2 g L$^{-1}$day$^{-1}$, using food waste as a substrate nutrient source.

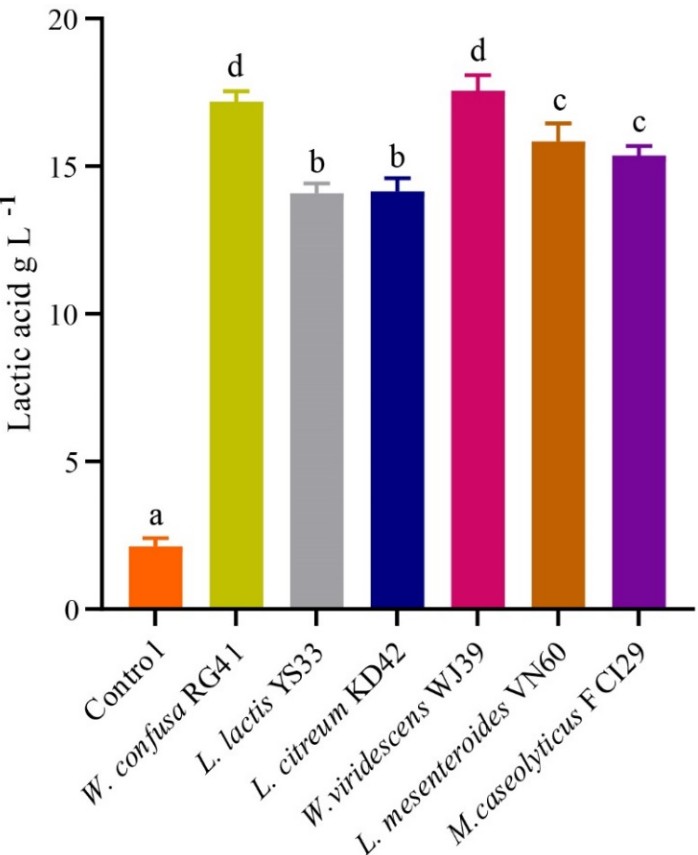

**Figure 5.** LA production potential of the bacteria at 24 h. The different letters denote a significant difference in the amount of LA production according to Duncan's test ($p < 0.05$).

Out of the six isolates (Figure 3), *W. viridescens* strain WJ39 (Figure 5) was selected as the strain with the highest potential for lactic acid production, and genome analysis was performed. The *W. viridescens* strain WJ39 features a 1.54 Mb genome length with a GC content of 41%. The genome assembly of *W. viridescens* WJ39 consisted of 16 contigs with an N50 of 215,217 and 13 scaffolds with an N50 of 501,242. The diagram was generated using the CGView Server. It shows the predicted genomic features and composition for *W. viridescens* WJ39. Progressing radially outward are GC-skew, GC content, rRNAs (Pink), tRNAs (red), and the coding (sense)/non-coding genes. The clusters of orthologous groups (COG) functional classification of coding genes is represented in different colors (Figure 6). Phylogenetic analysis (Figure 7) based on the 16S rRNA sequence revealed that the *W. viridescens* strain WJ39 isolates had a close relationship with *Weissella viridescens* strain NRIC1536. Based on the draft genome sequence of *W. viridescens* WJ39, a *hicD* gene known to catalyze the conversion of pyruvate to D-lactate was revealed in the genome.

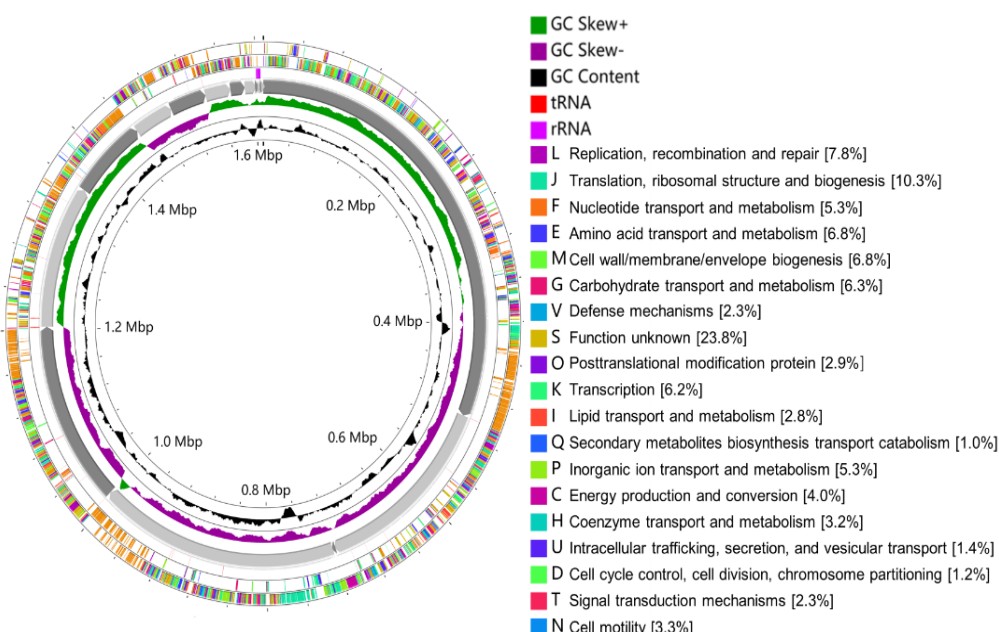

**Figure 6.** The diagram was generated using the CGView Server. It shows the predicted genomic features and composition for *W. viridescens* WJ39. From the inside to the outside are GC content, GC-skew, 16 contigs, RNA (rRNAs (purple), tRNAs (red)), the non-coding genes (antisense), coding genes (sense). The clusters of orthologous groups (COG) functional classification of coding genes is represented in different colors.

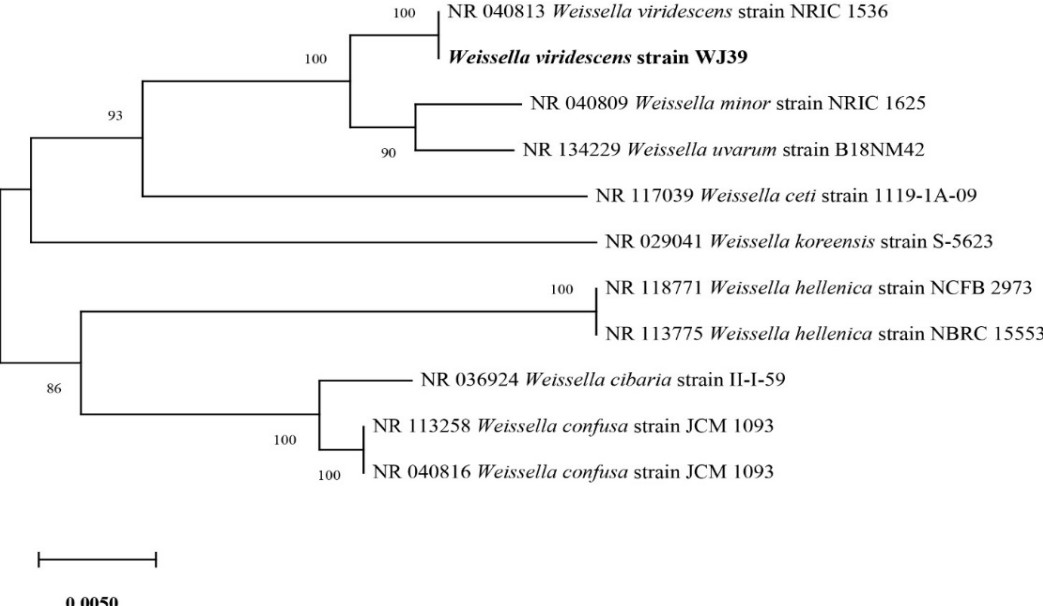

**Figure 7.** Phylogenetic tree based on the sequencing results of the 16S rRNA genes extracted from the genome shows the close relatives of *W. viridescens* strain WJ39.

## 4. Discussion

The results demonstrated that food waste is a source of beneficial strains and that the bacterial strains showed LA production potential. In recent studies, researchers described that lactic acid bacteria (LAB) are found in wide sources [19]. For example, from traditional pickles, 15 isolates of LAB were obtained, such as *Enterococcus faecalis*, *Lactococcus lactis*, *Lactobacillus plantarum*, *Pediococcus pentosaceus*, *Leuconostoc mesenteroides*, and *Enterococcus* spp., in MRS medium at 30 °C for 48 h [40]. LAB belonging to the *Enterococcus*, *Fructobacillus*,

*Lactobacillus*, *Lactococcus*, *Leuconostoc*, and *Weissella* genera were isolated from fruits of papaya, guava (pink and yellow varieties), custard apple, passion fruit, medlar, mulberry, khaki, and fig, and flowers of passion fruit, papaya medlar, and custard apple, in MRS medium [17]. *Bacillus* strains were isolated from Korean fermented soybean [58]. *Lactobacilli* were isolated from the oral cavity, dairy products, and feces using MRS agar at 28 °C [59]. These findings indicate that LA bacteria can be found in various habitats.

Previous studies showed different growth rates in different LAB species. The growth rate for *L. lactis* ranged from 0.23 h$^{-1}$ to 0.30 h$^{-1}$ in the MRS medium within 24 h [59]. *Weissella hellenica*, grown at 10 °C, shows delayed growth at 4 °C and does not grow at 37 °C [60]. We found that all isolated bacteria grew well within 24 h, with *W. viridescens* WJ39 having the highest growth rate (0.80 h$^{-1}$) and *Macrococcus caseolyticus* FCI29 (0.57 h$^{-1}$) having the lowest in the MRS medium. *Lactobacillus fermentum*, *Lactobacillus reuteri*, *Lactobacillus acidophilus*, and *Lactobacillus plantarum* showed growth rates of 0.62 h$^{-1}$, 0.41 h$^{-1}$, 0.38 h$^{-1}$ and 0.19 h$^{-1}$ in malt medium, 0.43 h$^{-1}$, 0.20 h$^{-1}$, 0.13 h$^{-1}$ and 0.18 h$^{-1}$ in barley, and 0.53 h$^{-1}$, 0.23 h$^{-1}$, 0.13 h$^{-1}$, and 0.15 h$^{-1}$ in wheat, respectively [61]. These results suggest that the different species have different growth rates under the same media, which can explain the various growth rates of the isolated bacteria observed in this study. Previously, it was reported that the growth rate at the exponential phase for *Lactobacillus plantarum* isolated in breast milk was 0.77 h$^{-1}$ at the optimal temperature [62]. *Lactobacillus* spp. were incubated in MRS and semisolid fermentation SSF (flours: wheat, maize, barley, and rye) media. The maximum growth rate (μmax) in the SSF medium varied from 0.21 to 0.37 h$^{-1}$, whereas the μmax of bacteria grown in the MRS medium did not exceed 0.23 h$^{-1}$ [63]. These results suggest that the medium composition could affect the growth rates of bacterial strains. In addition, *Lactobacillus acidophilus* CRL2074, *Lactobacillus amylovorus* CRL2115, *Lactobacillus mucosae* CRL2069, and *Lactobacillus rhamnosus* CRL2084 were isolated from a feedlot cattle environment, and their growth rates were assessed in MRS medium and found to be 0.48 h$^{-1}$, 0.45 h$^{-1}$, 0.51 h$^{-1}$, and 0.62 h$^{-1}$, respectively [44]. In our case, we investigated the growth rates of isolated bacteria using same medium (MRS), and all isolated bacteria scored growth rates generally higher than other findings.

Food waste and municipal sludge processed with *Lactobacillus rhamnosus* AW3 were used for LA production, and a maximum yield of 28.4 g L$^{-1}$ was produced after 48 h [64]. LA (27 g L$^{-1}$) was obtained in 48 h from food waste by *Lactobacillus delbrueckii* and by the addition of a commercial amylolytic enzyme (a mixture of amyloglucosidase, α-amylase, and protease) [65]. In addition, bacteria belonging to the *Bacillus* genus were identified as potent L-LA producers with yields ranging from 0.6–0.49 g L$^{-1}$ [30]. Fermented rice beverages with *Lactobacillus brevis* CCM 1815, *Lactobacillus fermentum* CCM 7192, *Lactobacillus plantarum* CCM 7039, and *Bifidobacterium longum* CCM 4990 were used to generate LA, with the highest LA amount (2.14 ± 0.03 g L$^{-1}$) produced by *Lactobacillus plantarum* CCM 7039 [66]. LA concentrations of 17.75 g L$^{-1}$ by *Lactobacillus plantarum* DSM and 12.72 g L$^{-1}$ by *Lactobacillus manihotivorans* DSM in MRS medium were reported in a previous study [46]. Our results showed a difference in LA production among the isolated bacteria (Figure 5), where the highest amount of LA production was 17.56 g L$^{-1}$ by *Weissella viridescens* WJ39, and the lowest was 14.09 g L$^{-1}$ by *L. lactis* YS33 in MRS medium within 24 h. These results suggest that the amount of LA production depends on the bacterial species and also the cultivation conditions. Our findings revealed that the strains from food waste have the potential for LA production, and food waste could be a substrate of lactic acid production.

## 5. Conclusions

A huge amount of food waste is produced annually. Poor food waste management causes negative environmental consequences that affect humans, including foul gas emissions and diseases. Various studies are proposed for food waste treatments using environmentally friendly methods to produce organic compounds for future utilization. This study provides operational information about bacterial isolation from food waste and applications related to bacterial strains. The study isolated and identified six bacterial

strains from food waste: *W. viridescens* WJ39, *L. lactis* YS33, *L. citreum* KD42, *W. confusa* RG41, *M. caseolyticus* FCI29, and *L. mesenteroides* VN60. All six bacteria exhibited the potential for lactic acid production in an MRS medium at 28 °C. The *W. viridescens* WJ39 strain demonstrated the highest growth rate (0.80 h$^{-1}$) and the highest lactic acid production rate (17.56 g L$^{-1}$day$^{-1}$). Overall, the results suggest that food waste could be considered as a substrate for beneficial bacterial strains for LA production.

**Author Contributions:** Conceptualization, B.L.; data curation, H.P.U., F.L. and A.T.; formal analysis, H.P.U., X.W. and Y.T.; investigation, H.P.U., Y.T. and B.L.; methodology, H.P.U. and X.W.; resources, H.P.U. and F.L.; software, H.P.U. and F.L; supervision, B.L.; visualization, A.T.; writing—original draft, H.P.U.; writing—review and editing, H.P.U. and B.L. All authors have read and agreed to the published version of the manuscript.

**Funding:** This research was funded by a Postdoc funding from Department of Human Resources and Social Security of Hebei Province, grant number B2021005008.

**Institutional Review Board Statement:** Not applicable.

**Informed Consent Statement:** Not applicable.

**Data Availability Statement:** The 16S rRNA gene sequences of the six isolates (VN60, FCI29, RG41, YS33, KD42, and WJ39) were deposited in NCBI under accession numbers ON387328, ON387327, ON387326, ON387331, ON387330, and ON387329, respectively.

**Acknowledgments:** Uwamahoro Henriette Peace acknowledge financial support from the UCAS Scholarship for Masters Students.

**Conflicts of Interest:** The authors declare no conflict of interest. The funders had no role in the design of the study; in the collection, analyses, or interpretation of data; in the writing of the manuscript; or in the decision to publish the results.

**Appendix A**

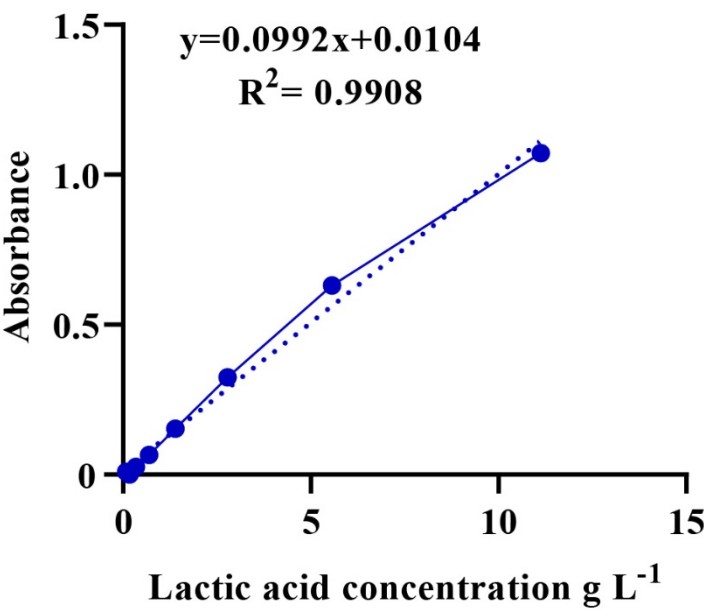

**Figure A1.** Calibration curve showing the absorbance and its corresponding concentration.

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
