# Peer review of "An Assessment of the Lactic Acid-Producing Potential of Bacterial Strains Isolated from Food Waste"

_2036-7481, doi:10.3390/microbiolres13020022_

Round 1

Reviewer 1 Report

The paper of Uwamahoro and colleagues reported the isolation of six different lactic-acid producer strains from food waste. Overall, the manuscript is well written and with an interesting focus, anyway without addressing some crucial points before publication it could remain poor of novelty. Specifically, there are some points that must be carefully addressed:

  • Why the draft genome of viridescens WJ39 is poorly descripted? This information could be the added value of the manuscript, instead it is quickly described and without discussion.
  • The strains were isolated and tested in aerobic or anaerobic conditions?
  • The whole 16S sequence of the six isolates should be deposited in NCBI database
  • Which is the carbohydrates concentration of the food waste? Also, the sterilization by means autoclave could damage sugars molecules, most likely a different sterilization method would have been more appropriate.
  • Why only one strain ( viridescens WJ39) was tested for the capability of LA production from food waste? Moreover, the production of 2 g L-1 day-1 it is not explanatory of its potential if the quantity of initial available sugars is not known. Furthermore, a discussion about this value and the real application of this process from food waste should be added. The discussion section should also consider the applications of LA microorganisms using real substrates, such as food waste and others, to understand the reliability of the process from an industrial point of view.
  • To date the few articles describing bacterial isolates lack microscopic images that can help to recognize their morphology. If available, images taken at optical microscope (e.g. gram staining) would be very informative and could be added in figure 2.

Author Response

Dear reviewer 1, Thank you for kind comments and suggestions. Kindly see the file attached.

Reviewer 2 Report

Thank you for submitting the manuscript “Assessment of the lactic acid-producing potential of bacterial 2 strains isolated from food waste” to the journal microbiology research. The market and demand for lactic acid production has been increasing due to its wide range of applications in different industries. It is valuable to produce lactic acid using food waste by means of fermentation. This manuscript covers information about the potential of bacteria to produce it, however it needs some changes before final submission. The authors need to address the following suggestions,

Line 49-55: The sentence is too long and needs to be rephrased for easy reading and to improve the quality of this manuscript.

Line 61: “is” expected to expand at…

Line 74: retted cassava should be explained properly, and it will help the readers to easily understand what “retted” means!

Line 99-100: I don’t agree that the mentioned reason explains why the food is being wasted all around the world!

Line 110: while “producing” LA….

Line 116: Mention the geographical coordinates as an insert in this line using brackets

Line 155-156: The comprehension must be clear to have a flow in the manuscript. It’s confusing as the authors had initially mentioned the propagation of strains in MRS, followed by the sterilization of media. Rewrite this sentence and be clear with the sentence order.

Line 170: When it comes maximum room temperature, do the authors have any approximate value it falls within a range? Because most of the laboratories regulates certain temperature and a controlled airflow that varies from place to place.

Under the heading 2.7, Bacterial growth rate and growth curve, after line 199, it would be nice to see the formula of calculating the growth rate instead understanding from the description in words.

Change the numbering to 2.9 under the heading STATISTICS

Under figure 5, I don’t really understand the difference as well as the comparison from the different letters mentioned for statistical difference. Can the authors please explain?

Line 272: highest potential for lactic acid production

Line 307: and 0.15 h-1 in wheat, respectively. Delete the word “wheat” at the start so that it has consistency in reading.

Under References, there is no uniformity throughout.  For example, most of the journal titles were in italics and few were not. Say for the reference numbers 8, 36…  were not highlighted. Review the complete section and correct it.

Overall comments and questions:

  • Do the authors will conduct any advanced works like enhanced production of LA, separation of LA, testing out different fermentation conditions to improve the production? If yes, it could be better to add few lines to show the next steps of this study.
  • Did the authors test any experiments on impact of storage of food waste? Just wanted to know as few papers have some information on that.

It’s interesting to showcase the isolation and production of lactic acid from the bacterial strains from food waste since it’s a value added to the source. By addressing the above suggestions and comments, the overall quality of the paper may improve. Nice work authors and good luck.  

Author Response

Dear reviewer 2, Thank you for kind comments and suggestions. Kindly see the file attached. 

Thank you!

Reviewer 3 Report

I find it interesting to remove 'monomero' from the summary. Monomer is a repeating unit in the polymer molecule

Lines 77-79, reorganize the sentence, got confused

Line 118 - change the term "rotting" to some synonym

Line 132 - Were the colonies purified? by which technique, why?

In molecular identification, the primer used from a sequence of some lactobacillus? Explain

In item 2.4, why was the pH adjusted to 6.2 before autoclaving?

Can't Figure 1 be an appendix? It was weird in the methodology. Can't be improved? it is clear, but can it be played in another program?

Lines 220-223 What led you to choose the 6 strains over the others?

“These results suggest that the amount of LA pro- 330 duced depends on the bacterial species.”... But also the cultivation conditions, you tested with 24 h, for example, the reference used 48 h and got a higher concentration. Can emphasize this

Author Response

Dear reviewer 3, Thank you for kind comments and suggestions. Kindly see the file attached.

Yours sincerely,

Round 2

Reviewer 1 Report

The revised manuscript microbiolres-1697569 of Uwamahoro and colleagues presented the identification and description of six different lactic-acid producer strains from food waste. Authors have addressed reviewers questions and suggestions and this version responded adequately to the reviewers comments.